# A Systematic Review on the Impact of the Social Confinement on People with Autism Spectrum Disorder and Their Caregivers during the COVID-19 Pandemic

**DOI:** 10.3390/brainsci11111389

**Published:** 2021-10-22

**Authors:** Yurena Alonso-Esteban, María Fernanda López-Ramón, Verónica Moreno-Campos, Esperanza Navarro-Pardo, Francisco Alcantud-Marín

**Affiliations:** PSiDEHESO Research Team, University of Valencia, 46010 Valencia, Spain; Yurena.Alonso@uv.es (Y.A.-E.); M.Fernanda.Lopez@uv.es (M.F.L.-R.); Veronica.moreno@uv.es (V.M.-C.); Esperanza.Navarro@uv.es (E.N.-P.)

**Keywords:** autism spectrum disorders, COVID-19 pandemic, systematic analysis, stress, anxiety and emotional regulation

## Abstract

The COVID-19 pandemic had imposed a variety of containment measures on the general population for prolonged periods. Confinement has had, and still has, social, economic, educational, health, and psychological consequences on the entire population. Objective: In this article, a systematic search has been performed based on studies carried out since the beginning of the pandemic, regarding the impact of these containment measures on the autism spectrum disorder (ASD) population and their caregivers. Method: We consulted six databases (i.e., PubMed, Medline, Embase, Scopus, Web of Science, and Science Direct) and selected ten studies that met the inclusion criteria. The chosen studies have been classified according to their theoretical focus, methodology, and target population. Results: We found an increase in stress and a decrease in psychological well-being among individuals diagnosed with ASD (i.e., parents and caregivers). Additionally, in studies focused on children, youth, and adults with ASD diagnosis, the results are contradictory depending on variables such as age, ASD severity, or type of family structure. Conclusions: The results show that the consequences of quarantine and social confinement are quite contradictory and depend on variables such as age, ASD severity, and family features.

## 1. Introduction

Three months after the appearance of the first cases of COVID-19 (SARS-CoV-2) in Wuham (China), the World Health Organization (WHO) [1] recognized the severity and extent of the pandemic. Since then, most nations, forced by high rates of hospitalization and mortality, had stablished sanitary measures of confinement, restriction of movement and social contact affecting the general population, and quarantine measures for persons infected or at risk of transmitting the virus. The social, political, and economic consequences of these measures are relevant and we considered that they require assessment in all of their dimensions. From a psychological point of view, research has already provided evidence of experienced stress increase in the general population [2,3,4,5,6,7] and in COVID-19 patients and health professionals [8,9,10,11]. Emotional disturbances, anxiety, and depression and their consequences [5,12,13,14,15] were also evaluated in the medium and long term. In addition, it is noted that some disorders such as autism spectrum disorder (ASD) may be a risk factor for infection and disease [16,17]. It is also noted that patients with COVID-19 and ASD plus intellectual disability (ID) showed longer and more intense hospital stays [18] and therefore represent a high-risk population who should have access to vaccines as a priority [19]. Added to the risk of contagion, the stress generated by the pandemic situation, uncertainty, isolation, and interruption of treatments showed that the ASD population and patients with mental health disorders can be more vulnerable and may present an increase in the intensity and frequency of comorbid symptoms (especially those related to anxiety and depression) [20,21]. For instance, anxiety and depression disorders are the most common mental disorders, [22] and often coexist [23] since both involve emotional dysregulation. In this sense, it seems reasonable to think that individuals suffering from anxiety disorders prior to the pandemic period are more vulnerable to experiencing further psychological stress in the pandemic context [24]. Family caregivers also seem more vulnerable when coping with (in addition to their own confinement and the stress produced by the pandemic) the care of persons with the disorder. Additionally, children, youth population, and adults with neurodevelopmental disorders (and in particular, those with autism spectrum disorders (ASD) [25]) can be considered more vulnerable as their treatments supports and routines have been discontinued [26].

One of the characteristics of ASD is the variability of symptoms and severity levels, as well as its comorbidity with other disorders [27]. Anxiety disorders (AD) are very common among people who develop ASD, shown by resistance to change [28], intolerance to uncertainty [29], emotional deregulation [30], and executive dysfunction [31] among other reasons [32]. In this line of argumentation, emotional regulation can be defined as the ability to modulate emotional response to stimuli in the direction of producing adaptive and socially appropriate behavior [33]. In the case of people with ASD, these responses can take the form of irritability, poor anger management, tantrums, self-injury, aggression, and mood deregulation [34] with serious consequences for caregivers as well [35]. Up to 84% of children with ASD experience some degree of anxiety [36]. Some empirical results, moreover, pointed out that the high-functioning ASD population (or those with less severe symptoms and without intellectual disability) have more frequent and intense anxiety disorders [24].

During the first months of the pandemic, the general population has had to assimilate a lot of public health information and apply sometimes contradictory recommendations and behavioral restrictions, generating a lot of uncertainty. In the ASD population, this situation was observed to be experienced with discomfort, emotional distress, or other anxiety symptoms [37,38]. Additionally, confinement, daily routine changes, and the suspension or reduction of their usual treatments and supports have had a negative impact on ASD individuals and their families [39], increasing stress and isolation as already documented in numerous scientific reports [40,41,42].

ASD patient’s relatives usually receive multiple kinds of support (i.e., therapeutic, educational) to cope with children’s needs [43]. In the new COVID-19 context, many families have seen their income cut as a result of the economic crisis associated with the pandemic [44,45], thus increasing the level of uncertainty, helplessness, anxiety, and depression among its members and increasing the number of cases of suicidal ideation and suicide attempts [46]. In a recent study carried out with ASD parents in the USA by Simons Powering Autism Research (SPARK) [47] during the first few days of the pandemic, it was reported that 84% of school services or therapies and 52% of clinical services were interrupted. In other regions (e.g., the European Union, China, South Korea, or Japan), the containment and closure of all non-essential activities has been the general tendency, with the subsequent interruption of therapeutic centers’ activities and support services consequently impacting families’ quality of life (e.g., in ASD patients and their families). On some occasions, medical, psychological, educational, and online services have been developed or reused in order to alleviate the effect of isolation. However, this solution has not been generalized and even additional stress has been reported with their use due to inadequate (in the case of those reused) or untrained caregivers [48].

In recent months, the number of publications based on the COVID-19 impact have increased. In this sense, the main objective of this work is to perform a systematic analysis of the research conducted which reflected the impact of the restrictive measures applied during the current COVID-19 pandemic on the population with ASD and their families with the aim of recommending the implementation of social measures, health, and educational programs to alleviate possible dysfunctions.

## 2. Methods

This study followed the guidelines of PRISMA (Preferred Reporting Items for Systematic reviews and Meta-Analyses) [49] for the methodological structure of search and screening methods.

### 2.1. Search Strategy

We performed a systematic search in several databases (e.g., PubMed, Medline, Embase, Scopus, Web of Science, and Science Direct) including research studies from March 2020 to date. We did the main searches on the 15th of march of 2021. In all of the searches performed, the following terms were used for the selection: (1) (COVID-19 or SARS-CoV-(2) AND (ASD OR Autism). In order not to lose information during the process of analyzing the documentation, we programmed an alarm in Google to retrieve the articles that will be published from 15 March until the end of the writing of this article. Figure 1 shows the flow chart selection process of the sample of articles analyzed.

### 2.2. Study Selection and Eligibility Criteria

As shown in Figure 1, step 4 reviewed the titles and abstracts of each publication to check their relevance and step 5 accessed the full-text articles to check their eligibility after the initial review. We excluded studies that: (1) were not peer-reviewed (rejecting opinion papers, general recommendations, or editorials); (2) were not available in full text; (3) were not focused on the ASD population exclusively; and (4) review and theorical review. The selected studies had to meet the criteria of (a) research articles (b) assessing the impact of psychological social restraints during the pandemic period (c) applied to the ASD population (in any age cohort) and their parents or caregivers.

## 3. Results

The initial search (1–6 steps) yielded fourteen studies that met the inclusion criteria. In addition (step 7), we included three additional studies identified from March to June 2021 through google alarm. Appendix A shows the most relevant data of the 17 studies (step 8) reviewed. It should be noted that, due to the pandemic situation, all studies consulted used online resources for data collection and/or pre-existing data in previous cohort studies. The results were organized according to whether the focus of the study was on physical activity psychological stress on parents or impact on the behavior of ASD children, young people, and adults.

### 3.1. Physical Activity in ASD Diagnosed Individual during Confinement and Quarantine

The reduction of physical activity in the general population due to confinement or quarantine by COVID has been studied [50]. For instance, Exentürk [51], assessed a ten parent sample of ASD diagnosed children using a qualitative approach wherein the parents were asked about the amount of physical activity performed by their children during COVID-19 confinements or quarantines. As a result, they organized the information in three blocks: possible benefits (i.e., health, social, psychological and prevention of addiction to technological tools), barriers to physical activity (i.e., occupations of family members, safety concerns, insufficient distance learning infrastructure, anxiety about disruption of routine and lack of knowledge), and solutions to increase physical activity (i.e., family education, expert support and resource support). In a similar line of argumentation, García et al. [52] performed a study on nine adolescents with ASD and applied a survey about physical activity time, screen exposure time, and sleep duration before and during confinement due to the pandemic. The results showed a decrease in physical activity versus an increase in screen exposure time (not observing differences in sleep regulation). The development of interventions that promote physical activity in ASD populations during COVID-19 confinements or quarantines is recommended. Pfeiffer et al. [53], analyzed mobility changes on a single case research, using mobility patterns acquired prior to the pandemic period and afterwards, by recollecting data provided by the GPS device on their cell phones. Social participation and mobility of all participants (6 young people aged 21–27 years) decreased significantly in both essential and non-essential activities.

### 3.2. Stress and Distress in Parents during Confinement and Quarantine

There are many studies that focused on parental stress or other emotional well-being alterations on parents caused by the change in routine as a result of confinement. In this line of argumentation, Colizzi et al. [54] developed an online survey to analyze the impact of the pandemic on the well-being and needs expressed by 529 caregivers of people with ASD in Italy. They found that most caregivers (i.e., 93.9%) experienced greater than normal difficulties in organizing their children’s activities during confinement, with behavioral problems being common. Manning et al. [55] conducted an online survey in Michigan (USA) of 471 families to determine the influence of confinement on families’ quality of life. Three main groups of stressors were identified as those related to the caregiver (i.e., financial, health, responsibility for children, etc.); those related to the person with ASD as a stressor (i.e., behavioral problems, isolation, changes in routines, etc.); and the absence of services as a stressor (i.e., absence from school, concern about follow-up care, etc.). It is worth noting that there is a direct relationship between the severity of the ASD symptoms and the stress level of the caregivers. Carers demanded respite services during COVID-19, and the higher the severity of the cared-for person’s symptoms, the higher the demand for respite services.

Furthermore, Alhuzimi [56] conducted a study comparing pre- and post-confinement measures in a group of 150 Saudi Arabian parents with children under 18 years of age diagnosed with ASD. They used standardized measures (i.e., such as PSI-SF, Parenting Stress Index short form [57] for parental stress measurement and GHQ [58] for measuring parents emotional well-being). Authors highlighted the differential impact in parents’ stress and emotional well-being and considered the severity of symptoms, gender, and age of the child (i.e., during COVID-19 pandemic, parental stress scores were increased and the emotional well-being scores were decreased).

Althiabi [59], applied a battery of standard ad hoc and online-built tests on a sample of 211 parents of children with ASD in a Saudi Arabian sample. They assessed the following variables: attitude, family impact (FIQ, [60]); anxiety HADS (hospital anxiety and depression scale [61]); mental health status, perception of mental health care GHQ (general health questionnaire [58]) and perceived need for mental health care through an ad hoc questionnaire. Two measures were taken, before and after pandemic confinement, with significant differences between the two assessment moments. They concluded that confinement had a negative impact on parents’ anxiety and health status.

On the other hand, Wang et al. [62] performed a similar the study with a larger sample. They recruited a total of 1764 ASD children’s parents and 4962 typically developing children’s parents (SD). Participants completed a descriptive survey of the pandemic impact: CD-RISC resilience (Connor-Davidson resilience scale [63], SCSQ coping styles [64], SAS anxiety (self-rating anxiety scale [65]) and SDS depression (self-rating depression Scale [66]. In general, ASD children’s parents showed lower resilience levels and positive coping and used more negative coping strategies than SD children’s parents. ASD children’s parents also developed more symptoms of anxiety and depression than SD children’s parents.

Meral [67], developed a study in Istanbul, Turkey, on 32 parents of children with ASD in which distress (Brief Family Distress Scale [68]), quality of life (Family Quality of Life Scale [69]), and family happiness were probed using a mixed design of qualitative and quantitative questions. The results suggested that working parents were distressed by the economic and security consequences of job loss. In contrast, middle-class families highlighted positive aspects such as having more time for parent-child interaction. The most negative aspect remarked by all participants was the closure of therapeutic and educational services for their children.

### 3.3. Impact on Children, Youth and Adults with ASD

Amorim et al. [26] conducted a descriptive study using an online survey of parents in Portugal. A total of 99 parents of children ranged from 6 to 12 years old (i.e., 43 parents had a child diagnosed with ASD and 56 with standard development –SD) participated. Parents reported more behavioral changes in ASD children than CD children (i.e., focusing on manifestations of anxiety, irritability, obsession, hostility, and impulsivity), but both groups reported a negative impact on learning or homework. Regarding emotional deregulation, the differences between both groups according to the informants were significant, this being greater in children with ASD. With respect to caregivers, all of them showed higher levels of anxiety than those manifested in their children, these being higher in ASD parents.

Jeste et al. [70] performed a survey of over 800 caregivers of children with intellectual and developmental disabilities to assess the reduction in educational care that they experienced during the confinements and quarantines resulting from COVID-19 pandemic restrictions. The obtained results indicated that more than 70% of those surveyed caregivers (i.e., both inside and outside the United States) experienced a decrease in educational or therapeutic services. For instance, in 56% of the US surveyed cases, they reported changes in intervention modalities, while outside the US only 32% reported so. In general, the changes in therapeutic modality service also involved a reduction in the number of sessions. As a consequence of this quality care decrease, caregivers’ burden increased and consequently compromised health and psychological children and their caregivers’ well-being.

Mutluer et al. [71] conducted a comparative study (i.e., before and after COVID-19 pandemic) on a sample of 87 patients from Koc University Hospital (Istanbul, Turkey) aged 3 to 29 years. Diagnostic information and measures from the aberrant behavior checklist (ABC [72]) were available in the pre-COVID period. A post-confinement assessment was performed, and they found that the ABC scores differed significantly with increasing severity. Similarly, the Pittsburgh sleep quality index (PSQI [73]) was administered, showing a worsening of the quality and number of hours of sleep and an increase in hypersensitivity. Regarding the caregivers, the BAI (Beck anxiety inventory [74]) was used and indicated a direct correlation between the caregivers’ high anxiety scores and the severity of ASD symptoms observed.

Berard et al. [75] studied the behavioral effects caused by confinement in France, assessed on a ASD young participants sample (i.e., ELENA study [76]) and compared it with a sample assessed prior to COVID-19 confinement. For it, they used an ad hoc online questionnaire consisting of five domains (i.e., nutrition, sleep, challenging behaviors, communicative skills, and stereotyped behaviors), evaluated using a three-point Likert scale (i.e., no change, better, or worse). They evaluated 239 families with children and youth, living together during confinement, with parents reporting worsening and deteriorating challenging behaviors and a relative worsening of sleep and repetitive and stereotyped behaviors. The results they found suggested that children and youth with ASD had been affected by confinement to the same extent as their peers with neuro-typical development including psychological symptoms as fear, irritability, and sleep problems.

Lugo-Marín et al. [77] conducted a study on the effects of containment measures and social distancing using a set of online administered tests (e.g., including standardized tests such as the CBCL (Child Behavior Checklist [78]), the SCL-90-R (Symptom Checklist 90 Revised [79]) and ad hoc questionnaires on the consequences of confinement and social estrangement. The study was part of the population monitored in the PAITEA program (Program of Integral Attention to Autistic Spectrum Disorder) of the Vall d’Hebron Hospital in Barcelona (Spain). A total of 100 participants were recruited and grouped according to age (i.e., as children and adolescents up to 17 years and 11 months of age, or adults 18 years and over) and the scores pre-post and during confinement were compared. As a conclusion, they found that differences in stress rates were lower in adults with ASD diagnosis than the ones registered on their caregivers. This result can be related with the difficulties in ASD participants in emotions recognition. Additionally, as a result of the drastic decrease in their social interactions associated with measures of social distancing and confinement, ASD participants could have experienced a reduction of social interaction demands that can be correlated with the observed reduced social anxiety.

Mumbardó-Adam et al. [80] studied the impact of COVID-19 confinement in 47 families with children (i.e., between 2 and 17 years old) diagnosed with ASD through an online survey. ASD parents reported the rise of new behaviors such as: greater participation in family routines, being more communicative with parents, greater levels of autonomy in self-care, etc., while few developed new stereotyped behaviors. Approximately 50% reported having observed changes in their children´s emotional status but in both directions (i.e., on the one hand, 40% reported that children were happier and calmer, while in others, 23.4% reported that they were more irritable and less often; 8.5% were observed to be sadder and depressed).

White et al. [81], studied ASD parents that used phone-assistance services during confinement. The results indicated that emotional regulation deterioration was the main factor triggering children´s stress, while they observed an increase on parent’s stress (i.e., because of the increasing ASD children´s demands of support new routines). This study evaluates the use of telecare as an alternative to personal assistance. The results indicated that telecare assistance efficacy is limited by the social communication problems of people with ASD.

Siracusano et al., [82] conducted an observational study on a sample of 85 children (aged 2–18 years) recruited from the clinical database of the Child Psychiatry Unit of the University of Rome Tor Vergata Hospital. Having diagnostic information and pre-confinement status by COVID-19, they develop an observation with measures such as the ABAS-II [83], RBS-R [84], and CBCL [78]. Although the research conducted evidenced certain changes, these will need to be evaluated over the long term in order to conclude the effects of the pandemic on ASD symptoms, in this specific sample.

## 4. Discussion

The studies analyzed are mainly descriptive and, due to the limits of the pandemic, the questionnaires have been distributed through digital media. Although the strengths and weaknesses between online measures versus standard questionnaires (i.e., paper and pencil mode) have long been studied (e.g., [85,86]), in this case most studies have only used ad hoc questionnaires or surveys where the validity and reliability of the measuring instruments have not been considered. Four studies have adapted standardized questionnaires to assess psychological well-being or severity of symptoms [56,59,62,77] and only one reports a specific questionnaire of their own making [75]. Another general limitation is the sample size; by targeting a restricted population, and due to the circumstances of the pandemic, the samples analyzed were small (except for one of the studies) [62]. Regarding participant’s selection procedure, it is worth noting that in no case was it a random procedure (i.e., were generally convenience samples of participants recruited in previous programs (e.g., [75,77]).

We believe think that the analyzed studies can be subdivided into three main groups: (i) studies related with the reduction of physical activity as a result of confinement and the effect on ASD participants, (ii) studies focused on family impact of confinements and/or quarantines including a ASD diagnosed individual co-habiting, and (iii) studies dedicated to analyzing the effects of confinement and quarantine on ASD diagnosed participants.

Regarding the aforementioned first group of studies, we included the ones revised that studied the relationship between physical activity and emotional well-being in the general population (e.g., [87]) and in ASD participants (e.g., [88]). In general terms, they found that the effects of confinement have led to a decrease in physical activities in ASD participants and caused an increase in the amount screens hours exposure. In this sense, we think that future research will have to analyze if the changes persist and involve more consistent habits changes or were related only with confinement and quarantine periods. On the other hand, it will also be necessary to evaluate middle term and long-term efficacy of the so called “tele-assistance” or “tele-health” services that included programs and tutorials to induce ASD diagnosed participants to perform physical activity at home for compensating the decrease or lack of external physical activity [89].

A second group of studies referred to confinements and quarantines impact on ASD relatives and included the higher number of studies that we have revised in the present work. On general terms, the psychological stress increase experienced by parents during confinements and quarantines were reported in all the studies revised. The most frequently mentioned sources of stress are: (a) the pandemic situation itself (i.e., fear in relation to general health, fear of COVID-19 disease, contagion, etc.); (b) work-life balance problems (e.g., virtual work while caring for children); (c) economic difficulties resulting from the pandemic (e.g., loss of employment due to company closures); and (d) additional new obligations related to the introduction of special routines for their ASD-diagnosed children. On the other hand, in the studies revised wherein parents and children received any type of online educational or therapeutic services, it was found that parents reported a positive experience when receiving advice and support (e.g., [81]), except for some cases in which parents reported that it was perceived as another source of stress due to their lack of experience on conducing at home therapeutic activities or routines. In this sense, it is worth remarking that although guidelines and recommendations for confinements and quarantines periods have been published for ASD relatives (e.g., [90]), these suggestions have not been sufficiently divulged outside of the scientific community.

Regarding the effects of confinements and quarantines on ASD diagnosed participants, the revised results appeared to be in some way contradictory. That is, while in some cases stress, anxiety, and stereotyped and challenging behaviors due to difficulties in emotional regulation increased (e.g., [26,75]), in others the decrease of psychopathological symptoms (i.e., except for anxiety) was reported (e.g., [77]). Additionally, factors identified as having a higher risk of increased stress and psychological distress were: (a) the age of the participant with ASD; (b) the severity of the disorder and the amount of support needed; and (c) the family structure and the amount of previously acquired daily living skills or tasks [75]. As a common outcome, all reviewed studies comprising this third group reported that disruption of health support and treatment was the most relevant trigger that could be related to increased stress and distress.

As a general conclusion, while parents reported an increased level of stress due to COVID-19 confinements and quarantines, the results found in participants diagnosed with ASD are contradictory. One of the most relevant findings to highlight is the positive consequences valued by some participants diagnosed with ASD. This assessment could be due to the fact that social isolation and environmental invariance (by remaining in a familiar context) reduced social stress, producing an increase in the satisfaction and well-being of autistic individuals. It is possible that the negative consequences of confinements and quarantines for participants with ASD are observed precisely when they leave the confinement situation and are confronted again with everyday social activities. In this sense, we believe that more research is needed to understand the short- and long-term psychological effects of confinement and quarantine, especially on special needs populations and their families and caregivers.

However, we note the existence of a need for or the creation of transition programs that have as their main objective the reincorporation of participants diagnosed with ASD into regular activities after confinement. We also believed that for future research, it is relevant to develop digital adaptations that can be proposed as effective alternatives to face-to-face therapeutic sessions and that this can be carried out during periods of confinement [91]. To this end, the inclusion of online counselling, measurement, and monitoring programs for periods of confinement and quarantine should be present in all properly evaluated and validated educational and treatment centers. In addition, training programs for ASD parents should also include guidance on the use of online systems to deal with situations such as those experienced throughout 2020 and 2021.

## Figures and Tables

**Figure 1 brainsci-11-01389-f001:**
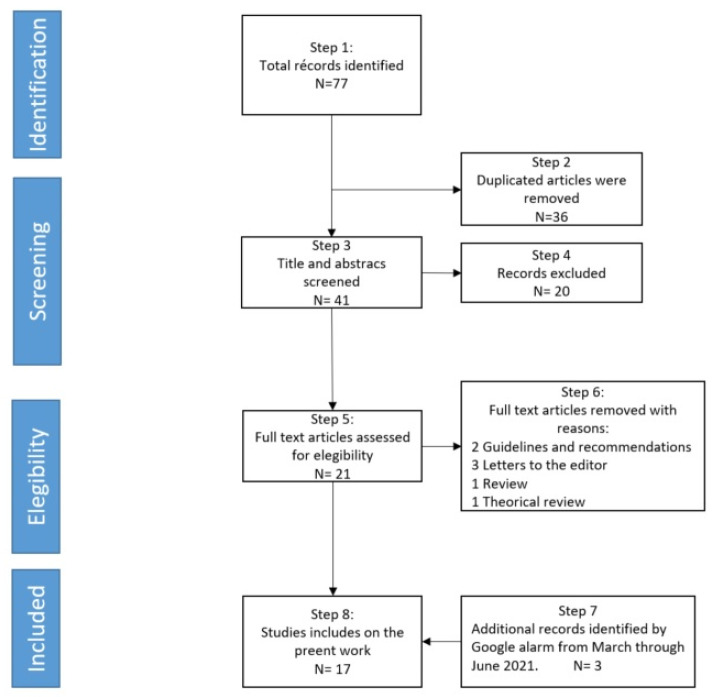
PRISMA review flow.

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
