# Peer review of "A Systematic Review on the Impact of the Social Confinement on People with Autism Spectrum Disorder and Their Caregivers during the COVID-19 Pandemic"

_brainsci, 2021, doi:10.3390/brainsci11111389_

Round 1
Reviewer 1 Report
This was an interesting topic to review, however, I did wonder if it was a little premature as articles investigating this topic may still be going through the review process. I would encourage the authors to revise the introduction to provide a clearer focus for the review. This would include explaining what is autism and why COVID restrictions may impact on individuals on the autism spectrum. The authors explained their exclusion criteria. Including an example of search terms and the inclusion criteria would also have added to the methods. When presenting the result of a systematic review, a mistake authors can make is reporting each study separately. This reads mre like a annotated bibliography. I would have like to have seen more synthesis of the results. PRISMA have a checklist for systematic reviews. http://prisma-statement.org/prismastatement/Checklist.aspx I would encourage the authors to work through this checklist. The impact of COVID on individuals on the autism spectrum and their families is an interesting topic.
Author Response
We are very grateful for the reviewer's comments and suggestions.
- a) We have made a critical reading and revised the wording of the whole manuscript trying to give more clarity both in the introduction and in the rest of the manuscript.
- b) The article is aimed at specialists and given that there is a limitation in the number of words, it was decided to omit explanations about ASD.
- c) The wording of the inclusion and exclusion criteria has been revised.
- d) As the reviewer acknowledges, this is the first review on this topic. Given the alarm generated by COVID-19 and the unpredictability of the pandemic, the studies carried out to date are of a low methodological level, with some exceptions. For this reason, we have preferred to be more descriptive. In the future and with more historical perspective, we will be able to review again the scientific production on the subject using the PRISMA checklist for systematic reviews.
Reviewer 2 Report
The timing of this review is quite perfect and the topic is more than interesting.
In this systematic review, the authors examined studies investigating the potential impact of confinement restrictions on patients with autism and their caregivers during the COVID-19 pandemic.
The results show that the consequences of the quarantine and the social confinement are quite contradictory and depend on variables such as age, ASD severity, and family features.
I have some concerns about the definition of this study. To me, this seems more a narrative review than a proper systematic review, even if the authors used the PRISMA guidelines.
Title:
- typo error in line 4 “itle”.
- I would rewrite the title this way: “A systematic review on the impact of the social confinement on people with Autism Spectrum Disorder and their caregivers during the COVID-19 pandemic”.
Abstract
Line 31-33: please rewrite the “Conclusions”; the sentence is quite confusing.
Introduction
Line 39: please use the extended name “World Health Organization” instead of the acronym.
Line 44: change “have” with “need”.
Line 54: remove “in particular”.
Line 58: change “deregulation” with “dysregulation”.
Line 59: remove period.
Line 63: “additionally”.
Line 65: change “especially” with “more”.
Line 80-83: please reformulate this paragraph.
Line 84: change “everyday” to “daily”.
Line 87: change “has been” in “already”.
Line 89: error “then”.
Line 94: remove the bracket “(“.
Lines 102-104: I suggest reformulating this sentence.
Methods
Since the authors chose to proceed with a systematic review, they should consider making more advanced analyses for the variables selected if possible.
To me, this review seems more a narrative review than a systematic review.
Line 117: “scientific if”???
The steps summarizing the study selection and the eligibility criteria should be clearer.
Please explicit the inclusion and the exclusion criteria like this: “Inclusion criteria were:1)…”.
Lines 134-135: authors should rewrite this sentence by linking it with the exclusion criteria n. 3.
I suggest including in this paragraph other elements about the selection of the studies reviewed. For instance, they might include some details about the variables selected that are explained in the “Results” paragraph, lines 140-142.
Results
General consideration: the authors should explicit in a better way the main findings of each study in this section. Sometimes they are not included at all.
Lines 140-142: consider removing this sentence by including it in the methods section (see the previous comment).
Table 1: please add a legend or a caption. I do not understand the use of the numbers in the brackets (1) identification etc… It is quite confusing; please change also the denomination of the first column “Identification”. Moreover, the authors should include another column summarizing the main findings of each study. Please add the reference number for each study reviewed. It would be very helpful for the readers.
Lines 146-147: please rewrite the sentence. Typo errors “were..were”.
Line 148-153: the authors should include also the results of the study reviewed.
Line 158-159: please rewrite the sentence.
Line 165: “Stress psychological distress”? Typo error?
Line 178-180: please explain better the findings of this study.
Discussion
At the beginning of this section, the authors should include an introductive paragraph summarizing some considerations about the main findings of the study reviewed, before explaining their limitations.
Line 307: consider removing the expression “We think”. The features of the studies reviewed should not be a random assumption of the authors. The subdivision in main groups should be the result of an objective and scientific analysis.
Line 320: please change “effectivity” in “efficacy” or “effectiveness”, since the authors are talking about the outcomes of a strategy.
Line 326-334: this paragraph is too long. The authors should use a clearer and synthetic structure.
Lines: 334-339: please rewrite the final part of this paragraph.
Lines 343: use a synonym for the word “effects”.
Lines 344: please change “That is” with “In fact”.
Line 348: “increased risk” for what? If the authors were referring to the effects of the restrictions, they should explicit this concept.
Moreover, the authors did not mention a possible explanation for these contradictory effects.
Please consider using synonyms for the term “additionally”.
Line 355-357: please rewrite this sentence. For example, the authors might say that: “about the effects of the confinement and quarantine restrictions on ASD patients, the result are quite contradictory”. Moreover, how the authors can certainly state that these controversial results are secondary to the children's age or to the severity of ASD symptoms? Please, explicit better this conclusion.
Line 357-358: please rewrite the sentence.
Line 369: the author should not include expressions such as “we think”. Please consider using more formal and objective language.
Author Response
We are grateful for the work and dedication of the reviewer. We have considered all the changes proposed:
Title:
The proposed title has been rectified and accepted.
Abstract
Line 31-33: the indicated sentence has been rewritten.
Introduction
Line 39: please use the extended name "World Health Organization" instead of the acronym.
OK
Line 44: change "have" to "need".
OK
Line 54: delete "in particular".
OK
Line 58: change "deregulation" to "deregulation".
OK
Line 59: remove period
OK.
Line 63: "additionally".
OK
Line 65: change "especially" with "more".
OK
Line 80-83: please reformulate this paragraph.
During the first months of the pandemic, the general population has had to assimilate a lot of public health information and apply sometimes contradictory recommendations and behavioral restrictions, generating a lot of uncertainty. In the ASD population, this situation was observed to be experienced with discomfort, emotional distress or other anxiety symptoms [37, 38].
OK
Line 84: change "everyday" to "daily".
OK
Line 87: change "has been" in "already".
OK
Line 89: error "then".
OK
Line 94: remove the bracket "(".
OK.
Lines 102-104: I suggest reformulating this sentence.
OK
Methods
As the reviewer acknowledges, this is the first review with this theme. Given the alarm generated by COVID-19 and the unpredictability of the pandemic, the studies carried out so far are of a low methodological level, with some exceptions. For this reason, we have preferred to be more descriptive. In the future and with more historical perspective, we will be able to review again the scientific production on the subject using the PRISMA checklist for systematic reviews.
Line 117: "scientific if"?????
OK
The wording of the inclusion and exclusion criteria has been revised in order to provide more clarity.
Lines 134-135: The wording has been revised and the reviewer's suggestion has been accepted.
Results
General consideration: Authors should make the main conclusions of each study more explicit in this section. Sometimes they are not included at all.
Lines 140-142:
OK
Table 1: the content of the table has been revised and changed.
Lines 146-147:
Revised paragraph and rectified
Lines 148-153:
Given the word count limitations for this type of article we have to be very concise and refer the interested reader to the original article.
Lines 158-159:
OK
Line 165:
Corrected
Line 178-180:
rectified
Discussion
The whole paragraph has been revised by introducing the changes suggested by the reviewer.
Round 2
Reviewer 1 Report
I appreciate the work the authors have done on revising their article. Althought they have revised the introduction this still requires a stronger and clearer argument for the study. I appreciate that there have been some studies on the impact of COVID on individuals on the autism spectrum, but I feel a systematic review is premature at this time. I would also encourage the authors to consider their language use when referring to individuals on the autism spectrum.